# The Impact of Internet Addiction on Mental Health: Exploring the Mediating Effects of Positive Psychological Capital in University Students

Girum Tareke Zewude [1,*] , Derib Gosim Bereded [1], Endris Abera [2], Goche Tegegne [1], Solomon Goraw [3] and Tesfaye Segon [4]

1   Department of Psychology, Wollo University, Dessie 1145, Ethiopia; derib.gosim@wu.edu.et (D.G.B.); goche.tegegne@wu.edu.et (G.T.)
2   Department of Curriculum and Instruction, Wollo University, Dessie 1145, Ethiopia; endris.abera@wu.edu.et
3   School of Law, Wollo University, Dessie 1145, Ethiopia; solomon.goraw@wu.edu.et
4   Department of Psychiatry, Mettu University, Mettu P.O. Box 318, Ethiopia; tesfayes721@gmail.com
*   Correspondence: girum.tareke@wu.edu.et

**Abstract: Introduction**: The widespread use of the internet has brought numerous benefits, but it has also raised concerns about its potential negative impact on mental health, particularly among university students. This study aims to investigate the relationship between internet addiction (IA) and mental health (MH) in university students, as well as explore the mediating effects of positive psychological capital (PsyCap) in this relationship. **Objective**: The main goal of this study was to examine the psychometric properties of the measures and to determine whether internet addiction could negatively predict university students' mental health, mediated through PsyCap. **Method**: A cross-sectional design with an inferential approach was employed to address this objective. The data were collected using the Psychological Capital Questionnaire (PCQ-24), Internet Addiction Scale (IAS), and Keyes' Mental Health Continuum-Short Form (MHC-SF). The total sample of this study comprised 850 students from two large public higher education institutions in Ethiopia, of whom 334 (39.3%) were female and 516 (60.7%) were male, with a mean age of 22.32 (SD = 4.04). Several analyses were performed to achieve the stated objectives, such as Cronbach's alpha and composite reliabilities, bivariate correlation, discriminant validity, common method biases, and structural equation modeling (confirmatory factor analysis, path analysis, and mediation analysis). Confirmatory factor analysis was conducted to test the construct validity of IAS, PCQ-24, and MHC-SF. Additionally, the mediating model was examined using structural equation modeling with the corrected biased bootstrap method. **Results**: The preliminary study results found that the construct validity of IAS, PCQ-24, and MHC-SF was excellent and appropriate. Furthermore, the findings demonstrate that internet addiction had a negative and direct effect on PsyCap and MH. Moreover, PsyCap fully mediated the relationship between IA and MH. Additionally, this study confirmed that all the scales exhibited strong internal consistency and good psychometric properties. **Conclusions**: This study contributes to a better understanding of the complex interplay between IA, PsyCap, and MH among university students, confirming previous findings. **Recommendation**: The findings, discussed in relation to the recent and relevant literature, will be valuable for practitioners and researchers aiming to improve mental health and reduce internet addiction by utilizing positive psychological resources as protective factors for university students.

**Keywords:** internet addiction; positive psychological resources; mental health; mediation analysis; university students

## 1. Introduction

Nowadays, internet addiction is a potential health-related problem at a global level. Internet addiction, characterized by excessive and compulsive internet use, has emerged

as a significant health issue affecting individuals worldwide [1]. It encompasses various problematic behaviors, including excessive use of social media, online gaming, and compulsive browsing [2,3]. Internet addiction has been associated with a range of negative consequences, including decreased academic and occupational performance, impaired interpersonal relationships, and mental health problems. The detrimental effects of internet addiction on mental health, including psychological, emotional, and social well-being, have led researchers to investigate the underlying mechanisms and factors associated with its development and impact [4–8]. A growing body of research examines the relationship between internet addiction, psychological capital, and mental health. For example, researcher [9] found a link between internet addiction and psychological health, suggesting that internet addiction can have implications for mental health that are relevant to overall psychological well-being. Researchers [8,9] stated there was a relationship between internet addictions, psychological capital (i.e., positive psychological resources), and mental health outcomes. The authors also found that excessive internet addiction was associated with lower psychological capital and poorer mental health. In addition, studies found that internet addiction negatively predicted mental health and a strong relationship was found between them [9–15]. Another study also by the researcher [12] in Turkey's adolescent sample found that internet addiction is a growing concern, and adolescents are the most vulnerable groups. Researchers [14] also investigated the relationship between internet addiction, psychological capital, and mental health outcomes. They found that higher levels of internet addiction were associated with lower psychological capital and poorer mental health. The researcher [6] investigated the relationship between internet addiction, moral values, and psychological well-being. They found that higher levels of internet addiction were associated with lower moral values and lower psychological well-being. Moreover, researchers [16–18] examined the relationship between social networking site (SNS) addiction, internet addiction, moral disengagement, and poor mental health outcomes. They found that SNS addiction and higher levels of internet addiction were associated with poorer mental health.

### 1.1. Benefits of Psychological Capital

Psychological capital (PsyCap) offers numerous benefits in the context of internet addiction and mental health. According to research [8], PsyCap refers to an individual's positive psychological resources, including self-efficacy, optimism, hope, and resilience. These resources have been found to play a vital role in buffering against the negative consequences of excessive internet use [19–22]. Firstly, PsyCap acts as a protective factor against internet addiction by equipping individuals with positive psychological resources. Individuals with higher levels of PsyCap are better equipped to cope with the challenges associated with internet addiction [8]. They are more likely to regulate their internet use, set boundaries, and seek support when needed. Research has shown that high levels of self-efficacy and resilience can help individuals in managing their internet use effectively [19–22].

Secondly, individuals with higher levels of psychological capital are more resilient in the face of setbacks [8]. They possess the ability to bounce back from negative experiences related to internet addiction, learn from their mistakes, and develop strategies to manage their internet use more effectively. This resilience helps them overcome the negative consequences of excessive internet use and maintain their well-being. Optimism and hope, which are integral to psychological capital, contribute to a positive mindset and motivation for recovery [13]. Research has indicated that optimism and hope foster a positive outlook that encourages individuals to address their internet addiction and work towards recovery [8]. These positive psychological states motivate individuals to make positive changes, seek help, and adopt healthier online behaviors.

Psychological capital also empowers individuals to exercise self-control and regulate their internet use. By enhancing self-efficacy and confidence [8], PsyCap enables individuals to make conscious choices about their online behavior and set limits. It empowers them to engage in activities that promote a healthier balance between their online and offline lives.

Furthermore, psychological capital contributes to improved mental well-being. Research has shown that internet addiction is associated with adverse mental health outcomes [2], but individuals with higher levels of PsyCap are less likely to experience symptoms of depression, anxiety, and reduced self-esteem [8]. These positive psychological resources act as a buffer against the negative effects of internet addiction on mental health.

PsyCap also offers the opportunity for individuals to develop and strengthen positive psychological resources. Through interventions and support, individuals can enhance their self-efficacy, resilience, hope, and optimism [17]. This development of positive resources contributes to improved mental well-being and healthier patterns of internet use.

The recognition of the benefits of psychological capital in relation to internet addiction and mental health has important educational and clinical implications [23]. Research has indicated that applying positive psychological resources can be an effective strategy for overcoming internet addiction and boosting mental well-being [24–29]. This understanding can guide the development of targeted interventions, policies, and educational programs aimed at promoting healthier and more balanced digital lifestyles.

### 1.2. Conceptual Framework

Internet Addiction, Psychological Capital, and Mental Health

The relationship between internet addiction, psychological capital (PsyCap), and mental health is an area that has received limited research attention. Internet addiction refers to excessive and compulsive internet use, resulting in negative consequences on various aspects of an individual's life, including their mental health and overall well-being [2,30,31]. Research has shown that internet addiction is associated with adverse mental health outcomes; psychological capital can act as a protective factor. Individuals with higher levels of psychological capital are better equipped to cope with the challenges associated with internet addiction. For example, high self-efficacy and resilience can help individuals regulate their internet use, set boundaries, and seek support when needed [23,32,33]. Optimism and hope can foster a positive mindset, encouraging individuals to address their addiction and work towards recovery [13]. A scientific model suggested by researcher [34] indicated that positive psychology interventions played a vital role in mental well-being and healthcare. In addition, internet addiction has been associated with increased symptoms of depression, anxiety, and reduced self-esteem, self-efficacy, hope, and optimism, which are factors that can undermine PsyCap [8,9]. Individuals with an internet addiction may experience lower resilience, optimism, hope, and self-efficacy levels, but further research is needed to establish a direct and indirect relationship.

### 1.3. Gaps in the Literature

Although there is growing research on the internet on (e.g., [6–8,11–23]) psychological capital [8,9,17,26] and mental health [24,25], the existing research has not addressed the issue of the mediating role of PsyCap between IA and MH among undergraduate university students and the protective role of positive psychological resources—particularly from the perspective of developing countries, which have witnessed an explosive growth in the population and the need of technology using internet. Motivated by this research opportunity, this study aims first to conduct the psychometric suitability of the three measures in the Ethiopian cultural context and examine the association of IA and MH mediated through PsyCap in an Ethiopian sample.

The existing literature on internet addiction (IA), psychological capital (PsyCap), and mental health (MH) has several notable gaps.

First, there is a lack of research focusing on the mediating role of PsyCap between IA and MH, particularly among undergraduate university students in developing countries with a significant growth in internet usage [17,26]. The existing studies have mainly focused on developed countries, leaving a gap in understanding the specific dynamics in different cultural contexts.

Second, there is a lack of validation of measurement scales in Ethiopian cultural context. The psychometric suitability of the Psychological Capital Questionnaire (PCQ-24) extended version, Internet Addiction Scale (IAS), and Keyes' Mental Health Continuum-Short Form (MHC-SF) has not been established in the Ethiopian context. Validating these scales is crucial for conducting reliable research and comparing findings across different cultural contexts.

Third, the need for empirical research on the relationships between IA, PsyCap, and MH. To date, no empirical research has examined the predictive role of IA on MH mediated through PsyCap. Conducting empirical research to determine the nature and strength of these relationships is essential for advancing knowledge in this field and informing interventions, policies, and educational programs [8,35].

Fourth, there is limited research specifically targeting undergraduate university students, who may be particularly vulnerable to IA and its impact on MH.

Fifth, there has been inadequate exploration of the educational and clinical implications of the relationships between IA, PsyCap, and MH, particularly in developing strategies to overcome internet addiction and enhance mental well-being.

Consequently, based on the above concrete empirical and theoretical evidence and the issue's importance, this research aims to address this subject by exploring the predictive role of internet addiction on mental health. Moreover, this study investigates the mediating effects of psychological capital (PsyCap) in this relationship based on the theoretical framework of the cognitive-behavioral model of internet addiction [30], the broaden-and-build theory of positive emotions [31], and the positive psychology theory [36]. Therefore, this study explored the impact of internet addiction on mental health [25] as a mediator of PsyCap [8,35]. This was completed using self-report measures of the Internet Addiction Scale (IAS; [22]), a Psychological Questionnaire (PCQ-24; [8]), and a Mental Health Continuum-Short Form (MHC-SF; [25]).

By addressing these research gaps, we can better understand the complex relationships between internet addiction, psychological capital, and mental health. This knowledge can guide the development of targeted interventions, policies, and educational programs to promote healthier and more balanced digital lifestyles [21]. Overall, to the best of our knowledge, except for the role of these comprehensive mediation models, the role of internet addiction on mental health mediated through positive psychological capital has not been examined anywhere in the world. Therefore, to fill this gap, empirical research must be conducted to determine the nature and strength of these relationships.

### 1.4. Research Questions and Hypotheses

Consequently, it is essential to note that the research on internet addiction, psychological capital, and mental health is still in its early stages, and further investigation is needed to establish more robust conclusions [23,32,33]. The relationship between these constructs is likely to be complex and influenced by various individual and contextual factors. Additionally, the direction of causality is not well-established, and it is possible that internet addiction can both influence and be influenced by psychological capital and mental health. Therefore, we propose the following testable research questions and hypotheses:

Research Questions

1. To what extent do the Amharic versions of the Internet Addiction Scale (IAS), the Psychological Capital Questionnaire (PCQ-24), and the Mental Health Continuum-Short Form (MHC-SF) exhibit high levels of reliability and validity?
2. Is there a negative relationship between Internet Addiction (IA) and Psychological Capital (PsyCap) and Mental health (MH) among undergraduate young students?
3. Does PsyCap positively predict MH among undergraduate young university students?
4. Does PsyCap mediate the relationship between IA and MH among undergraduate young university students?

Research Hypotheses

**Hypothesis 1**: *The Amharic versions of the IAS, the PCQ-24, and the MHC-SF will demonstrate high levels of reliability and validity.*

**Hypothesis 2**: *There is a negative relationship between IA and PsyCap and MH among undergraduate young students.*

**Hypothesis 3**: *PsyCap positively predicts MH among undergraduate young university students.*

**Hypothesis 4**: *PsyCap mediates the relationship between IA and MH among undergraduate young university students (see Figure 1).*

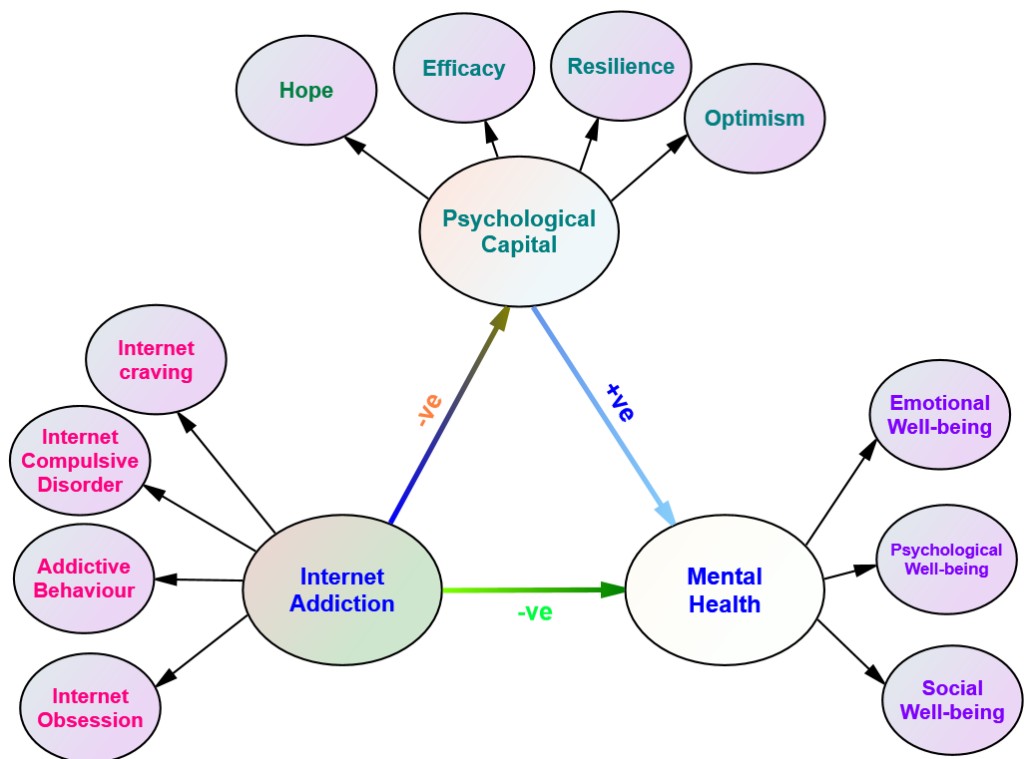

**Figure 1.** A Conceptual model of the IA on MH mediated by PsyCap. *Note:* **Direct Effect:** Internet Addiction → PsyCap Internet Addiction → Mental Health. Psychological Capital → Mental Health. **Indirect Effect:** Internet Addiction → PsyCap → Mental Health.

## 2. Methods

### 2.1. Research Design

The current study employed a quantitative research design with an associational and cross-sectional approach, deemed well-suited to achieving the stated objectives.

### 2.2. Sample and Sampling

This study was conducted in the Amhara Regional State of Ethiopia, specifically at two public universities. These universities were selected due to their proximity to the authors and prior experience with these sites. The sampling frame was divided into subsections based on specific characteristics, and a random sample was selected from each stratum. Initially, 889 university students were randomly selected to participate in the surveys. However, due to missing information, mistakes, or carelessness in data entry, 39 participants were excluded from the analysis, resulting in an effective response rate

of 95.6%. The researchers completed a breakdown of the sample based on gender (male and female) and batches (university years): Freshman, Sophomore and Senior. There were 516 male students, accounting for 60.7% of the sample, and 334 female students, accounting for 39.3% of the sample. The participants had a mean age of 22.32, with a standard deviation of 4.04.

In the second strata, the researchers further grouped the university years into three levels or batches. From the initial sample of 850 respondents (which represents 95.6% of the total sample), participants were stratified based on their year in the university and gender. The final sample breakdown for each university year was as follows: Freshman (1st year)—233 students (124 male and 109 female), Sophomore (2nd year)—260 students (160 male and 100 female), and Senior (>2nd year)—357 students (232 male and 125 female). These selections were made using a simple random sampling technique. The sample size in this study was determined based on established guidelines, where a small sample size is around 100, a medium size is approximately 150, and a large size is greater than 200. To ensure statistically stable estimates and minimize sampling errors, it is recommended to have a sample size of 200 or more [38]. Therefore, the sample size in this study was determined taking into account the power of the test and following the guidelines of [38].

*2.3. Instruments*

2.3.1. Socio-Demographic Information

Undergraduate students self-reported their gender, batch, age, and university.

2.3.2. Internet Addiction Scale (IAS)

IAS developed by [22] aimed to assess an individual's excessive and compulsive internet use that interferes with daily functioning using a 17-item scale. The IAS was a seven-item response option ranging from Very Strongly disagree (1) to Very Strongly agree (7). IAS has four main dimensions with very good psychometric properties: Internet craving (IC; Five Items; $\alpha = 0.771$, CR = 0.836), Internet compulsive disorder (ICD; Four items; $\alpha = 0.776$, CR = 0.857), Addictive behavior (AB; Four Items; $\alpha = 0.741$, CR = 0.802), Internet obsession (IO; Four Items; $\alpha = 0.663$, CR = 0.862) and the total scale ($\alpha = 0.878$).

2.3.3. Psychological Capital Questionnaire (PCQ-24)

The PCQ-24 scale is a 6-point Likert scale ranging from 1 (strongly disagree) to 6 (strongly agree) developed by [8] to assess respondents' positive psychological resources which includes four subscales. Each of these subscales consists of six items: (1) hope ($\alpha = 0.80$, 0.72, 0.75, 0.76), (2) self-efficacy ($\alpha = 0.84$, 0.75, 0.85, 0.75), (3) optimism ($\alpha = 0.76$, 0.74, 0.69, 0.79), (4) resilience ($\alpha = 0.71$, 0.66, 0.71, 0.72) and the overall PsyCap scale ($\alpha = 0.89$, 0.88, 0.89, 0.89). The PCQ-24 has been previously published, tested and validated in different culture and languages. For example, in Lithuania [39], South Africa [40], France [41], Brazil [42].

2.3.4. Mental Health Continuum-Short Form [MHC-SF]

MHC-SF is the most widely used instrument designed to measure the status of the mental health of an adolescent. The Keyes' Mental Health Continuum-Short Form [MHC-SF] was used to measure the mental health of the participants [25]. The MHC-SF instrument was used to assess 'the frequency of happiness, social belongingness to a community and managing the psychological functioning of daily life' [25]. Three major clusters comprising social, emotional, and psychological well-being were used to assess the healthy functioning of the adolescents' mental health. The scale developed by [24] covered three dimensions: social well-being (5 items; $\alpha = 0.74$), emotional well-being (3 items; $\alpha = 0.85$), and psychological well-being (6 items; $\alpha = 0.84$) and comprised 14 items [25]. The overall scale of MHC-SF reliability was $\alpha = 0.91$. Respondents rate each item on a 7-point Likert scale, ranging from 1 (Very strongly disagree) to 7 (Very strongly agree). This scale possessed excellent construct validity and reliability [25].

*2.4. Statistical Data Analysis*

To analyze the data, this study utilized several statistical software packages, including the Statistical Package for the Social Sciences (SPSS) version 29 and Smart PLS 4.1.0. Two key aspects of this study were the evaluation of psychometric properties and the examination of mediation effects. To ensure the suitability of the measurement instruments, the researchers followed a rigorous scientific procedure, incorporating various methods for assessment. However, the cross-cultural validation process encountered challenges, particularly with questionnaire translation and measurement of other instruments [37,43,44]. To address these challenges, the researchers adhered to the validation guidelines proposed by [44]. This involved a step-by-step process, including initial translation, synthesis, back translation, expert review, and administration and validation. Additionally, the researchers followed the recommendations of [45,46] for instrument validation. Overall, the validation process and the findings related to mediation were derived from four distinct procedures.

i. *Normality distribution*: The absence of multicollinearity was confirmed by examining the correlation matrices among the constructs, which should be less than 0.90, and by verifying the assumption of normality. Outliers of the constructs were also examined following the procedures of [46–50]. Values of $\leq 2$ or $\leq 4$ for skewness or kurtosis, respectively, indicate the normal data distribution [51,52].

ii. *Multi-collinearity*: To identify any potential issues with multi-collinearity in the data, the researchers used VIF (Variance Inflation Factor) and tolerance measures, as suggested by [46,47]. Additionally, the Harman single-factor test was used to assess the bias caused by common method variance.

iii. *Reliability assessment*: The researchers evaluated the internal consistency of the sub-scales using two measures: CR (Composite Reliability) and Cronbach's alpha coefficient. Excellent internal consistency was indicated by values above 0.90, while values between 0.80 and 0.90 were considered good, and values between 0.70 and 0.80 were deemed acceptable [48,49].

iv. *Convergent and Discriminant Validity*: Convergent and discriminant validity were evaluated to confirm the construct validity of the measurement instruments. Convergent validity was assessed using the Average Variance Extracted (AVE), where values exceeding 0.5 indicate satisfactory convergent validity. Discriminant validity was examined by comparing the Maximum Shared Variance (MSV) with the AVE values of the variables. Adequate discriminant validity was established when the MSV value was lower than the AVE value, and the squared correlation between sub-constructs was lower than the AVE value, indicating acceptable discriminant validity [46].

v. *Confirmatory factor analysis (CFA) and mediation analysis*: Confirmatory factor analysis (CFA) and mediation analysis were conducted to examine the factorial validity of the measurement and structural models [46–50]. Maximum likelihood estimation was utilized to identify the measurement and structural relationships within the proposed model. The goodness-of-fit of the models was assessed using several indices, including the normed chi-square ($\chi^2/\mathrm{df}$), Tucker Lewis Index (TLI), Comparative Fit Index (CFI), Standardized Root Mean Residual (SRMR), and Root Mean Squared Error of Approximation (RMSEA). An excellent and satisfactory fit of the models is typically indicated by $\chi^2/\mathrm{df}$ values below 3 or 5, RMSEA and SRMR values below 0.08 and 0.01, and TLI and CFI values above 0.95 and 0.90, respectively [53]. These indices provide information about how well the proposed models align with the observed data, with lower $\chi^2/\mathrm{df}$, RMSEA, and SRMR values and higher TLI and CFI values indicating a better fit [53]. To examine indirect effects, the researchers calculated 95% bias-corrected and accelerated confidence intervals using the bootstrap method with 5000 resamples. Through various sophisticated approaches, the researchers successfully addressed the challenges associated with psychometric assessment and mediation analysis, providing valuable insights into the variables and relationships examined in this study.

### 2.5. Procedures of the Studies

2.5.1. Adaption, Translation, and Validation of the Measures

Cross-cultural validation is a method for selection, adaptation, and validation of instruments designed in one culture for use in research in another country with culturally different populations, using the following priorities [37]: (a) Instruments that have been extensively tested and found psychometrically sound in one culture but have not been tested and determined meaningfully applicable for use in another culture; (b) Instruments that have high face validity but require further psychometric testing in another culture. It has often been applied in social science research in which self-reporting measures or measurement tools, usually questionnaires adapted for use in other languages other than the original [37].

The instruments used in this study were initially developed for other cultures; namely, the *Psychological Capital Questionnaire (PCQ-24)* Internet Addiction Scale (IAS) developed by [22] for the Indian culture, the *Psychological Capital Questionnaire (PCQ-24)* by [7] and the Keyes' Mental Health Continuum-Short Form [MHC-SF] by [25] instruments were developed for English-speaking samples for the US cultural context. Crucially, the portability of the IAS, PCQ-24, and MHC-SF to a culturally diverse and predominantly non-North American and non-Asian environment should be investigated before inferences derived from the measures are used with confidence in the Ethiopian context. Therefore, cross-cultural validation is necessary [45]. As a result, this study followed the proposed guideline of [43,44] Hence, we performed the following stages based on the proposed guideline of [44] regarding cross-cultural validation. *Stage 1: Initial translation/Forward Translation*. The first stage in the adaptation process is the forward translation method. The forward translation of the English version of the IAS, PCQ-24, and MHC-SF was performed by two experienced bi-lingual translators, one of whom was an English Language Training expert and the other a psychologist. *Stage 2: Synthesis of the translation*. In this stage, we selected two translators; one is a bi-lingual language expert but uninformed, and the other is a psychology professor who is informed about the purpose of the questionnaire. *Stage 3: Back Translation*. The translator then translates the instruments back into English. *Stage 4: Expert Review*. In this stage, both translators meet and check the cross-cultural equivalence of the instrument. There was no discrepancy between the original and the translated version found. *Stage 5. Administration of the instrument/Validation*. The final stage of the adaptation process is the pretest and validation processes.

Mediation analysis is the second goal of this study after ensuring the validation and the suitability of measures in the study context. Mediation analysis is defined as the process in which an indirect effect, where the effect of the independent variable on the dependent variable goes through a mediator [46]. Mediation analysis is needed because the relations may be modified or informed by adding a third variable in the research design. In addition, the relationships between psychological variables are often more complicated to make inferences in simple bi-variate correlation [46]. According to [43,46] also noted that the mediation model must address three main questions at the end of this study: (a) Is the indirect effect significant? (b) Is the mediated main effect significant? and (c) If both mediated main effect and indirect effect are significant, what is their relative strength and significant? As a result, to test mediation analysis, four processes were carried out: (i) confirmatory factor analysis, (ii) examination of measurement and structural model test, (iii) Path analysis, (iv) structural equation modeling.

2.5.2. Ethics of this Study

The data collection process followed the guidelines set by the American Psychological Association. Participation in this study was entirely voluntary, and ethical procedures outlined in the Helsinki Declaration, including 21 CFR 50 (Protection of Human Subjects) and 21 CFR 56 (Institutional Review Boards), were strictly adhered to. The researchers ensured the participants' confidentiality and anonymity regarding their participation and

data. Additionally, the university, IRB provided a letter of ethical approval affirming the appropriateness of the data collection procedures for this study.

## 3. Results

### *3.1. Results of Preliminary Analysis*

#### 3.1.1. Descriptive Statistics, Skewness, and Kurtosis

Table 1 provides an overview of the descriptive statistics, including means, standard deviations, skewness, and kurtosis as indicators of distribution normality [48,51,52]. Where the data had a skewness of 2 or a kurtosis of 4, the data were considered to be regularly distributed [51,52]. All the constructs in this study had normal distribution, as seen by the study's skewness values, which ranged from −0.282 to −1.128, and kurtosis scores, which ranged from −0.040 to 1.580.

**Table 1.** Descriptive statistics, kurtosis, and skewness.

| Variables | Min | Max | Mean | Std. Dev | Skewness | Kurtosis |
|---|---|---|---|---|---|---|
| Internet craving | 5.000 | 25.000 | 20.032 | 4.87852 | −1.128 | 0.772 |
| Internet compulsive disorder | 4.000 | 20.000 | 14.719 | 4.201 | −0.745 | −0.202 |
| Addictive behavior | 4.000 | 20.000 | 15.528 | 3.870 | −0.962 | 0.368 |
| Internet obsession | 4.000 | 20.000 | 14.358 | 4.749 | −0.618 | −0.719 |
| Internet addiction | 17.000 | 85.000 | 64.636 | 13.720 | −1.080 | 1.580 |
| Hope | 6.000 | 36.000 | 23.696 | 7.663 | −0.831 | −0.040 |
| Efficacy | 6.000 | 36.000 | 21.701 | 8.300 | −0.282 | −0.770 |
| Resilience | 6.000 | 36.000 | 22.202 | 7.608 | −0.343 | −0.593 |
| Optimism | 6.000 | 36.000 | 21.423 | 7.749 | −0.385 | −0.720 |
| PsyCap | 24.000 | 144.000 | 89.023 | 25.91 | −0.489 | 0.288 |
| Emotional well-being | 3.00 | 21.00 | 11.615 | 4.105 | −0.786 | −0.494 |
| Psychological well-being | 6.00 | 42.00 | 25.115 | 7.719 | −1.080 | 0.249 |
| Social well-being | 5.00 | 35.00 | 21.139 | 6.472 | −0.489 | 0.241 |
| Mental health | 14.00 | 98.00 | 57.869 | 16.574 | −0.815 | 0.944 |

Notes: Max = maximum; Min = minimum; PsyCap = psychological capital; Std. Dev = standard deviation.

#### 3.1.2. Multi-Collinearity

According to [46], if the tolerance values of predictor variables in a model are close to each other, it indicates the absence of an issue with multi-collinearity. Conversely, if the tolerance values are close to zero, it suggests the presence of multi-collinearity. The VIF statistic, which should ideally range from 0 to 5, with lower values being more desirable (approaching 0), is used to assess multi-collinearity. When the VIF score exceeds 5, it indicates that certain predictor variables are linear combinations of others [46,47]. In our study, the VIF was below 5, and the tolerance limits for each independent variable were greater than or equal to 0.01. Therefore, we concluded that the independent variables did not exhibit multi-collinearity issues based on the VIF and tolerance measures (see Table 2).

**Table 2.** Tolerance and VIF of multi-collinearity statistics of Internet Addiction and PsyCap on mental health.

| Model | Unstandardized Coefficients | Standardized Coefficients | t | Sig. | Collinearity Statistics | |
|---|---|---|---|---|---|---|
| | Beta | Beta | | | Tolerance | VIF |
| Internet addiction | −0.265 | −0.219 | −7.697 | 0.000 | 0.920 | 1.086 |
| PsyCap | 0.324 | 0.507 | 17.819 | 0.000 | 0.920 | 1.086 |

To determine if our study had common method bias, we conducted the Harman single-factor test. The results indicated that all constructs had a common method bias rate

of 36.49%, which was below the recommended fit requirements. As a result, we concluded that the study findings were not impacted by bias resulting from common method bias.

Table 3 provides an overview of the interrelationships among the variables in this study. To assess whether there is a relationship among independent and dependent variables or not, the Pearson correlation matrix was conducted following the guidelines outlined by [51,52]. The results indicated a negative correlation between IA with PsyCap (r = −0.282, $p < 0.01$) and MH (r = −0.362, $p < 0.01$). Conversely, a positive correlation was observed between PsyCap and MH (r = 0.569, $p < 0.01$). However demographic factors such as gender, age, and batch/years of study did not correlate with the three main constructs (see Table 3 for detailed results). Therefore, we did not explore socio-demographic factors further among the main constructs.

### 3.1.3. Reliability and Validity Evidence of the Main Variables

To test the first research hypotheses, we examined the construct validity, construct reliability, and internal consistency of the variables on undergraduate students in Ethiopian higher educational settings. Reliability scores above 0.90 indicate high reliability, scores between 0.80 and 0.90 suggest good reliability, and scores between 0.70 and 0.80 indicate adequate reliability [46,49,54]. The validity and reliability of the internet addiction, PsyCap and mental health aspects were evaluated. The results for the reliability coefficients of the Internet addiction scale (IAS) aspects were as follows: internet craving ($\alpha$ = 0.927, CR = 0.927), internet compulsive disorder ($\alpha$ = 0.904, CR = 0.904), addictive behavior ($\alpha$ = 0.883, CR = 0.883), and internet obsession ($\alpha$ = 0.926, CR = 0.926), all of which were excellent. The reliability of the PsyCap sub-dimensions were tested ($\alpha$ = 0.952, CR = 0.953), and the values for each of the HERO dimensions were as follows: Hope ($\alpha$ = 0.9586, CR = 0.958), Efficacy ($\alpha$ = 0.950, CR = 0.951), Resilience ($\alpha$ = 0.946; CR = 0.947), and Optimism ($\alpha$ = 0.957; CR = 0.957). The reliability coefficients were also found for mental health (MH) dimensions as follows: emotional well-being ($\alpha$ = 0.934, CR = 0.934), psychological well-being ($\alpha$ = 0.964; CR = 0.964), and social well-being ($\alpha$ = 0.959, CR = 0.960). This study demonstrated the high reliability of the four components of IA, the four dimensions of PsyCap and the three key core elements of the MH construct in Ethiopian educational settings supported the first hypothesis.

To assess the validity of these constructs, we examined their discriminant and convergent validity using the IAS, Psychological PCQ-24, and MHC-SF. Table 4 presents the AVE, MVE, and CR values for the sub-components of the study variables. It was found that all four constructs of IA, the four PsyCap elements, and the three dimensions of MH have good convergent validity (AVE > 0.05), indicating that the items are composed of fundamental components with reasonable correlation. Discriminant validity was evaluated by comparing the AVE values with the MSV values, and it was observed that the AVE values were higher than the MSV values, indicating acceptable discriminant validity. The AVE values for the sub-constructs of the IAS, PCQ-24, and the MHC-SF met the criteria for convergent and discriminant validity. Additionally, discriminant validity was assessed by comparing the AVE with squared inter-item correlations, and it was found that the AVE for all sub-constructs was higher than the squared correlation for each construct, indicating satisfactory discriminant validity. Overall, the IA, PsyCap, and MH constructs meet the standards for convergent and discriminant validity in Ethiopian higher education.

**Table 3.** Pearson correlations (r) among the socio-demographic factors and the predictor variables with the criterion variables (N = 850).

| | Variables | 1 | 2 | 3 | 4 | 5 | 6 | 7 | 8 | 9 | 10 | 11 | 12 | 13 | 14 | 15 | 16 | 17 |
|---|---|---|---|---|---|---|---|---|---|---|---|---|---|---|---|---|---|---|
| 1. | Sex | - | | | | | | | | | | | | | | | | |
| 2. | Age | −0.006 | - | | | | | | | | | | | | | | | |
| 3. | Batch | 0.096 ** | 00.020 | - | | | | | | | | | | | | | | |
| 4. | IC | −0.068 * | −0.013 | −0.006 | - | | | | | | | | | | | | | |
| 5. | ICD | −0.008 | −0.001 | −0.011 | 0.613 ** | - | | | | | | | | | | | | |
| 6. | AB | −0.047 | 0.021 | −0.012 | 0.540 ** | 0.436 ** | - | | | | | | | | | | | |
| 7. | IO | −0.053 | 0.016 | 0.014 | 0.433 ** | 0.459 ** | 0.305 ** | - | | | | | | | | | | |
| 8. | HO | −0.111 ** | −0.007 | −0.045 | −0.216 ** | −0.165 ** | −0.083 * | −0.162 ** | - | | | | | | | | | |
| 9. | EF | −0.018 | 0.024 | −0.026 | −0.271 ** | −0.133 ** | −0.222 ** | −0.135 ** | 0.750 ** | - | | | | | | | | |
| 10. | RE | −0.137 ** | 0.006 | 0.011 | −0.269 ** | −0.253 ** | −0.143 ** | −0.248 ** | o.598 ** | o.570 ** | - | | | | | | | |
| 11. | OP | −0.033 | −0.014 | 0.058 | −0.226 ** | −0.163 ** | −0.225 ** | 0.028 | 0.515 ** | 0.564 ** | 0.506 ** | - | | | | | | |
| 12. | EWB | 0.044 | −0.011 | 0.034 | −0.258 ** | −0.242 ** | −0.190 ** | −0.282 ** | 0.458 ** | 0.457 ** | 0.429 ** | 0.272 ** | - | | | | | |
| 13. | PWB | 0.054 | −0.028 | 0.041 | −0.274 ** | −0.262 ** | −0.179 ** | −0.298 ** | 0.479 ** | 0.456 ** | 0.453 ** | 0.293 ** | 0.673 ** | - | | | | |
| 14. | SWB | 0.001 | 0.018 | 0.057 | −0.286 ** | −0.237 ** | −0.208 ** | −0.285 ** | 0.536 ** | 0.499 ** | 0.476 ** | 0.294 ** | 0.715 ** | 0.758 ** | - | | | |
| 15. | IA | −0.058 | 0.007 | −0.004 | 0.845 ** | 0.806 ** | 0.713 ** | 0.727 ** | −0.207 ** | −0.247 ** | −0.299 ** | −0.184 ** | −0.317 ** | −0.331 ** | −0.332 ** | - | | |
| 16. | PSYCAP | −0.069 | 0.003 | −0.001 | −0.296 ** | −0.214 ** | −0.204 ** | −0.155 ** | 0.863 ** | 0.874 ** | 0.802 ** | 0.778 ** | 0.487 ** | 0.507 ** | 0.544 ** | −0.282 ** | - | |
| 17. | MH | 0.036 | −0.016 | 0.050 | −0.303 ** | −0.275 ** | −0.212 ** | −0.320 ** | 0.546 ** | 0.520 ** | 0.503 ** | 0.319 ** | 0.840 ** | 0.928 ** | 0.920 ** | −0.362 ** | 0.569 ** | - |

* and **. Correlation is significant at the 0.05 level and 0.01 level (2-tailed), respectively; AB = Addictive behavior, EF = Efficacy, EWB = Emotional well-being, HO = Hope, IA = Internet addiction, ICD = Internet compulsive disorder, IC = Internet craving, IO = Internet obsession, MH = Mental health, OP = Optimism, PsyCap = Psychological capital, PWB = Psychological well-being, RE = Resilience, SWB = Social well-being.

**Table 4.** Reliability and Validity Indices of the Study Variables (N = 850).

**Internet Addicton Scale (IAS)**

| Models | α (>0.70 *) | CR | AVE (>0.50 *) | MSV | Squared correlation | | | |
|---|---|---|---|---|---|---|---|---|
| | | | | | 1 | 2 | 3 | 4 |
| 1. Internet Craving | 0.883 | 0.883 | 0.658 | 0.44 | - | | | |
| 2. Internet Compulsive Disorder | 0.927 | 0.927 | 0.719 | 0.44 | 0.44 | - | | |
| 3. Addictive Behavior | 0.904 | 0.904 | 0.703 | 0.34 | 0.34 | 0.22 | - | |
| 4. Internet Obsession | 0.926 | 0.926 | 0.760 | 0.26 | 0.22 | 0.26 | 0.11 | - |

**Psychological Capital Questionnaire (PCQ-24)**

| Models | α (>0.70 *) | CR | AVE (>0.50 *) | MSV | Squared correlation | | | |
|---|---|---|---|---|---|---|---|---|
| | | | | | H | E | R | O |
| Hope (H) | 0.958 | 0.958 | 0.791 | 0.59 | - | | | |
| Efficacy (E) | 0.950 | 0.951 | 0.764 | 0.59 | 0.59 ** | - | | |
| Resilience (R) | 0.946 | 0.947 | 0.745 | 0.38 | 0.38 ** | 0.34 ** | - | |
| Optimism (O) | 0.957 | 0.957 | 0.788 | 0.34 | 0.29 ** | 0.34 ** | 0.26 ** | - |

**Mental Health Continuum-Short Form (MHC-SF)**

| Models | α | CR | AVE (>0.50 *) | MSV | Squared correlation | | |
|---|---|---|---|---|---|---|---|
| | | | | | EWB | PWB | SWB |
| Emotional Well-Being (EWB) | 0.934 | 0.934 | 0.825 | 0.62 | - | | |
| Psychological Well-Being (PWB) | 0.964 | 0.964 | 0.818 | 0.59 | 0.50 ** | - | |
| Social Well-Being (SWB) | 0.959 | 0.960 | 0.826 | 0.62 | 0.62 ** | 0.57 ** | - |

**Note**: ** indicated 0.01 level of significant value, * Indicates a global rule of thumb of an acceptable level of validity and reliability based on the recommendation of [47,48]. α = Cronbach's alpha; AVE = average variance extracted; CR = composite reliability; MSV = maximum shared variance.

### 3.1.4. Measurement and Structural Model

The measurement model (M4) consisted of 3 latent constructs and 11 indicators. Specifically, the Internet Addiction Scale (IAS) had four indicators (internet craving, internet compulsive disorder, addictive behavior and internet obsession), the Psychological Capital Questionnaire-24 (extended version; PCQ-24) had four indicators (hope, efficacy, resilience, and optimism) and the Mental Health Continuum-Short Form (MHC-SF) had three indicators (emotional well-being, psychological well-being and social well-being). The measurement model demonstrated a good fit based on the confirmatory factor analysis (CFA) results. For the IAS, the model fit indices were $\chi^2(113) = 364.80$, TLI = 0.973, CFI = 0.977, and RMSEA = 0.051 (95% CI = 0.045, 0.057) (see Figure 2). The PCQ-24 also showed an acceptable model fit with $\chi^2(246) = 2005$, TLI = 0.915, CFI = 0.924, and RMSEA = 0.092 (95% CI = 0.088, 0.096) (see Figure 3). The MHC-SF exhibited an excellent model fit with $\chi^2(74) = 234.75$, TLI = 0.986, CFI = 0.989, and RMSEA = 0.051 (95% CI = 0.043, 0.058) (see Figure 4). Overall, the measurement model for all scales demonstrated a good fit to the data with $\chi^2(1375) = 4383.90$, $p < 0.001$, TLI = 0.935, CFI = 0.940, and RMSEA = 0.051 (95% CI = 0.049, 0.052), indicating that the latent variables were accurately represented by their indicators. The structural model, which evaluated the relationships between the constructs, also showed a good fit to the data with $\chi^2(1416) = 4660$, $p = 0.001$, TLI = 0.932, CFI = 0.935, and RMSEA = 0.052 (95% CI = 0.050, 0.054). All factor loadings were significant and ranged between 0.73 and 0.94, $p = 0.001$), indicating that the indicators effectively captured the underlying latent variables (see Table 5). In summary, the measurement model

and structural model both exhibited a good fit to the data, indicating that the indicators accurately represented the latent constructs and the relationships between the constructs were well-supported by the data.

**Table 5.** Confirmatory Factor Analysis of Measurement and the Structural Models of the Constructs (N = 850).

| Models | Variables of this Study | Fitness of Indices Using Confirmatory Factorial Analysis of the Variables | | | | |
|--------|------------------------|---------|-----|-----|------|-------|
| | | $\chi^2$ | TLI | CFI | SRMR | RMSEA |
| Model 1 | Internet Addiction (see Figure 2) | 364.80 (113) ** | 0.973 | 0.977 | 0.031 | 0.051 |
| Model 2 | PsyCap (see Figure 3) | 2005.74 (246) ** | 0.915 | 0.924 | 0.046 | 0.092 |
| Model 3 | Mental Health (see Figure 4) | 234.75 (74) ** | 0.986 | 0.989 | 0.016 | 0.051 |
| Model 4 | Measurement Model | 4384 (1375) ** | 0.935 | 0.940 | 0.037 | 0.051 |
| | Structural Model | 4660 (1416) ** | 0.932 | 0.935 | 0.046 | 0.052 |
| | Rule of Thumb | | >0.90 | >0.90 | >0.08 | >1.00 |

Note: ** $p < 0.001$, $\chi^2$ = chi-squared, df = degrees of freedom, TLI = Tucker Lewis index, CFI = comparative fit index, RMSEA = root mean error square of approximation. Model 1: Internet Addiction of CFA model (see Figure 2); Model 2: PsyCap CFA Model (see Figure 3); Model 3: Mental health CFA model (see Figure 4); Model 4: Internet Addiction → PsyCap → Mental health (see Figure 5).

### 3.1.5. Mediation Testing Using Structural Equation Modeling (SEM)

The present study utilized structural equation modeling (SEM) with latent variables and the bootstrapping method to investigate the mediating role of Psychological Capital (PsyCap) in the relationship between internet addiction and mental health. Path analysis was employed to examine a mediation model and point estimates along with a 95% bootstrap confidence interval were used to determine the parameters. Mental health served as the outcome (dependent) variable, while internet addiction and PsyCap were the predictor (independent) variables. The standardized coefficients and 95% confidence intervals obtained through bootstrapping for the structural model are presented in Table 6 and Figure 5. Also, we examined the proportion of variance (i.e., R2) explained by the predictor variables to measure the correctness of the prediction obtained with the structured model. The findings indicated that internet addiction explains the variance of mental health (48.1%) and psychological capital (28.2%) of the data indicated a better fit for the model. In addition, the model accounts for 32.3% of the variance of psychological capital on mental health. All these confirmed that the model was acceptable, and internet addiction and PsyCap predicted mental health.

**Table 6.** Estimated effects of predictors on mental health, including both direct and indirect effects using a 95% biased corrected confidence interval (N = 850).

| Predictors | Outcome Variables | Bootstrap 95% CI | | | |
|-----------|-------------------|------|-----|-----|---------|
| | | Beta | LBC | UBC | *p*-Value |
| | Standardized Direct Effect | | | | |
| Internet addiction | PsyCap | −0.327 | −0.414 | −0.248 | 0.001 |
| Internet addiction | Mental Health | −0.211 | −0.277 | −0.140 | 0.003 |
| PsyCap | Mental Health | 0.595 | 0.533 | 0.658 | 0.001 |
| | Standardized Indirect Effect | | | | |
| Internet addiction → PsyCap → | Mental Health (Figure 5) | −0.195 | −0.252 | −0.146 | 0.001 |

**Note:** CI = confidence interval, LBC = lower bound, UBC = upper bound, PsyCap = psychological capital.

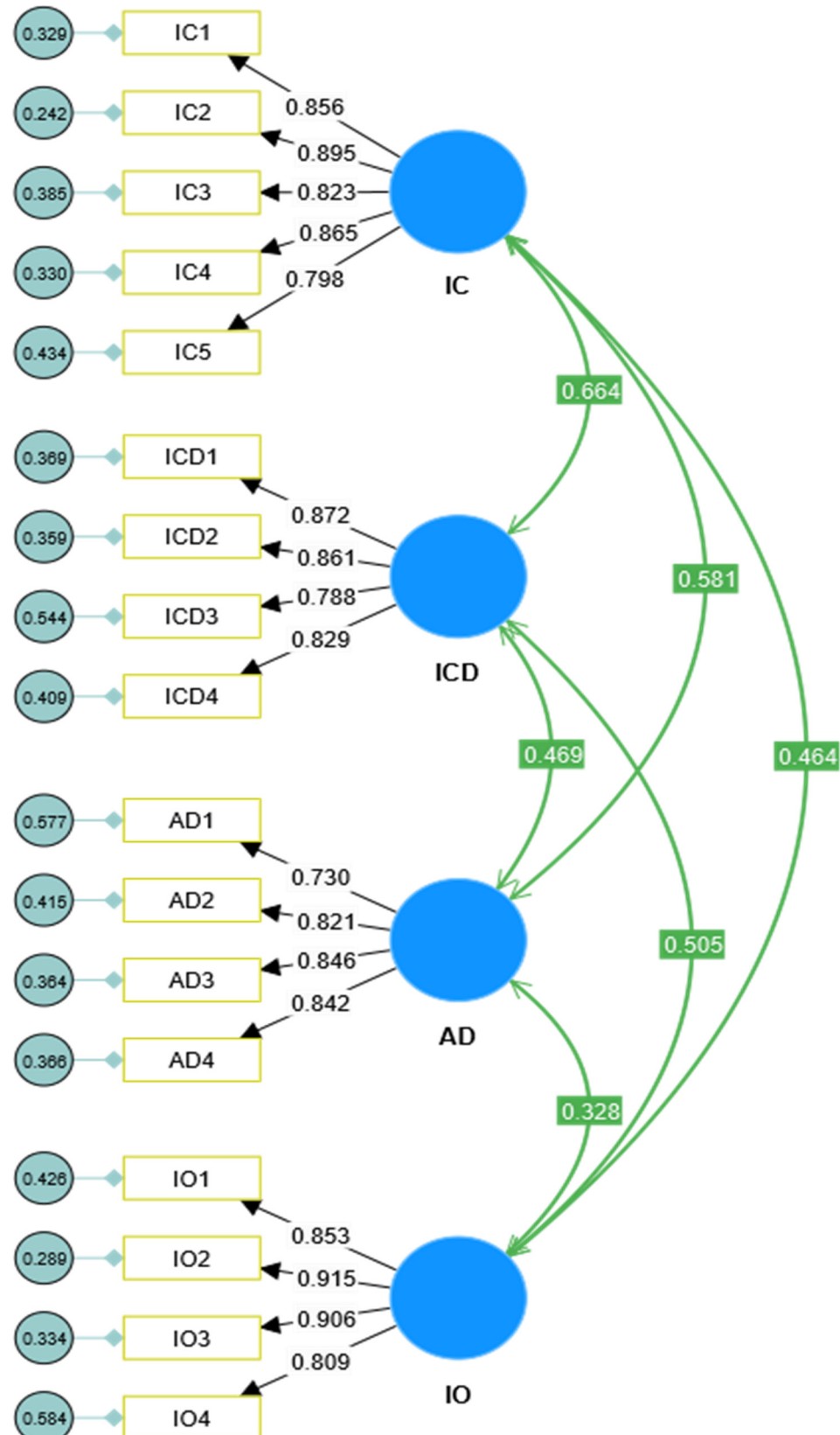

**Figure 2.** CFA Model of Internet Addiction Scale.

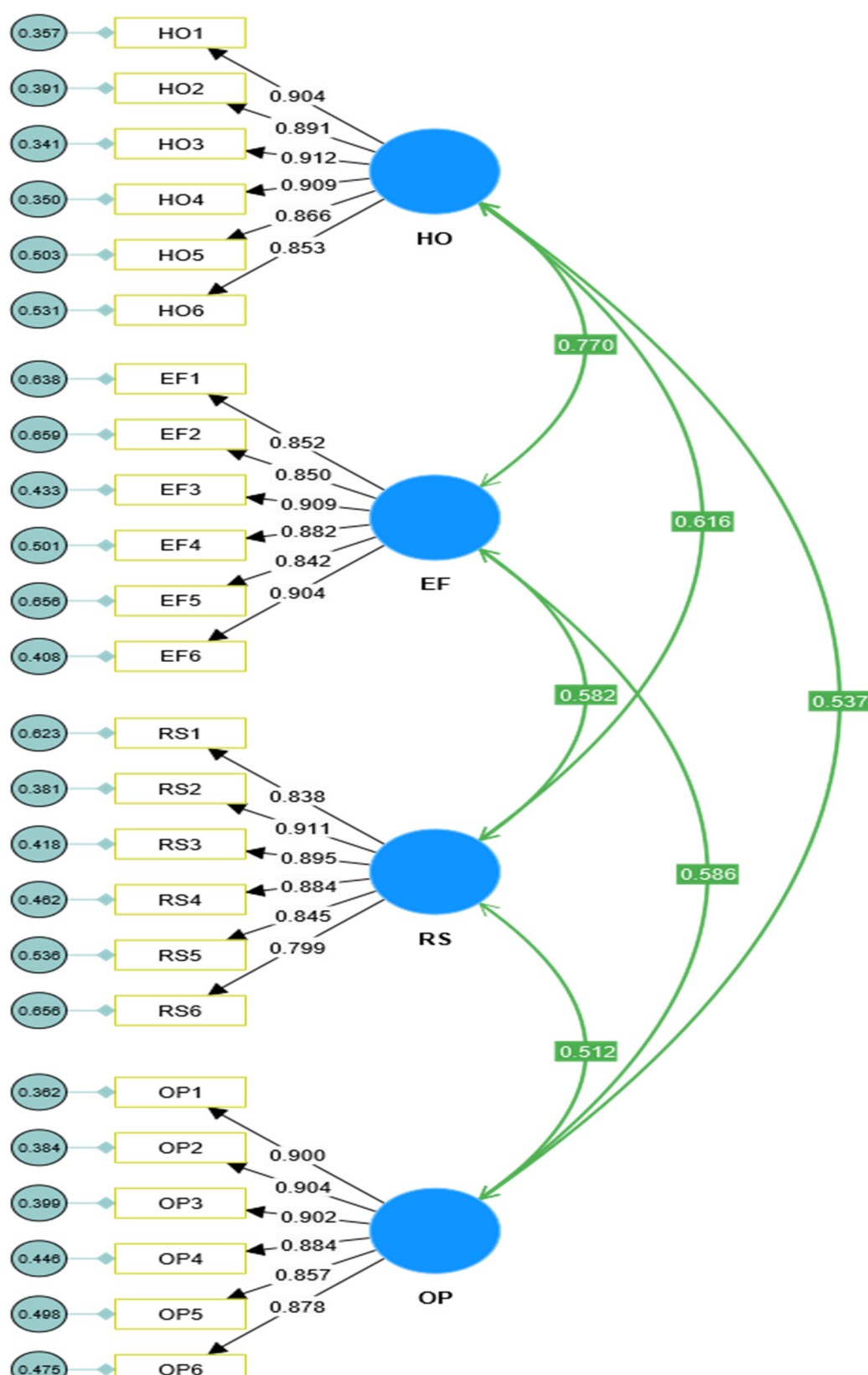

**Figure 3.** CFA Model of Psychological Capital Questionnaire.

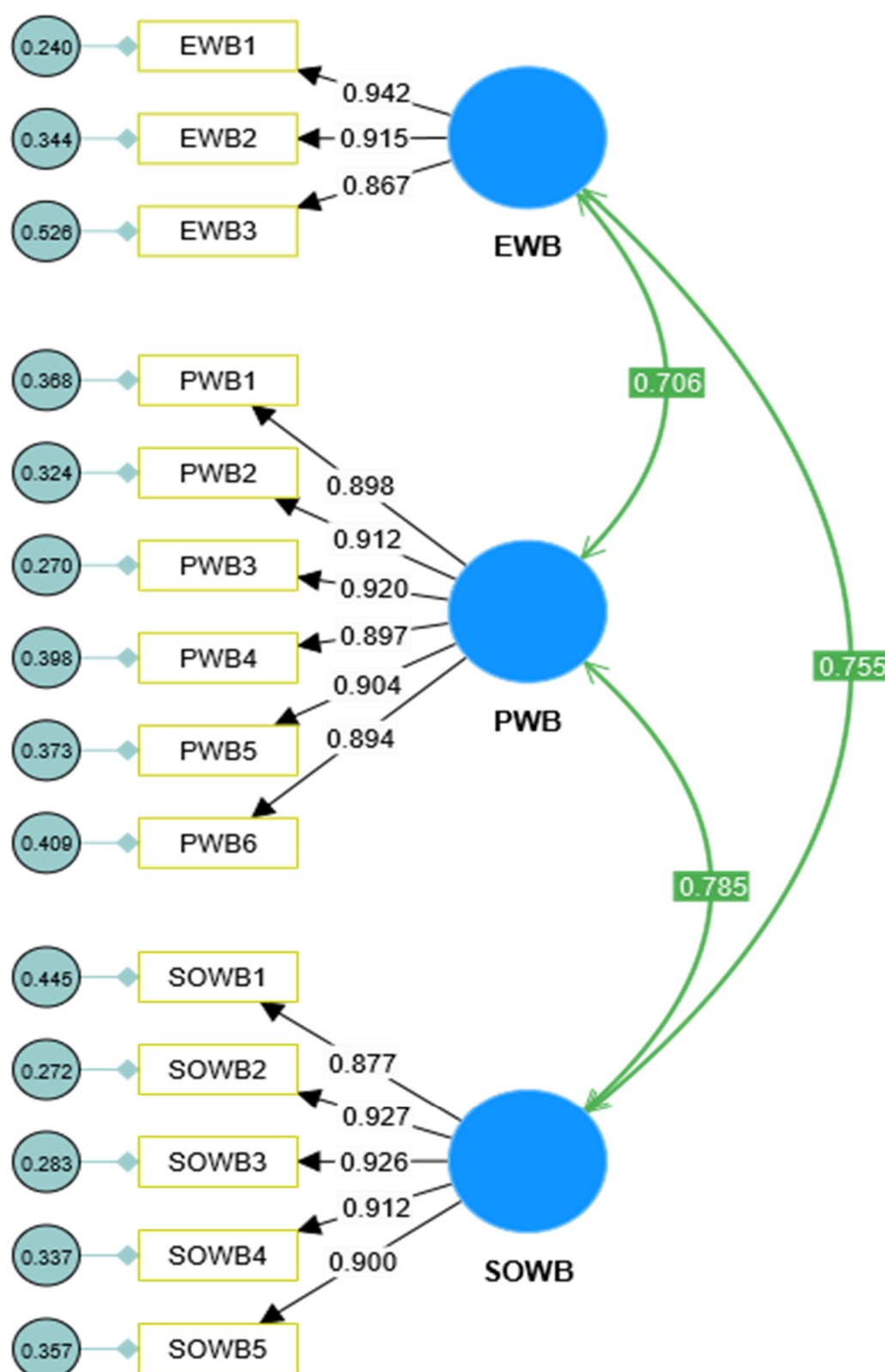

**Figure 4.** CFA Model of Mental Health Continuum-Short Form (MHC-SF).

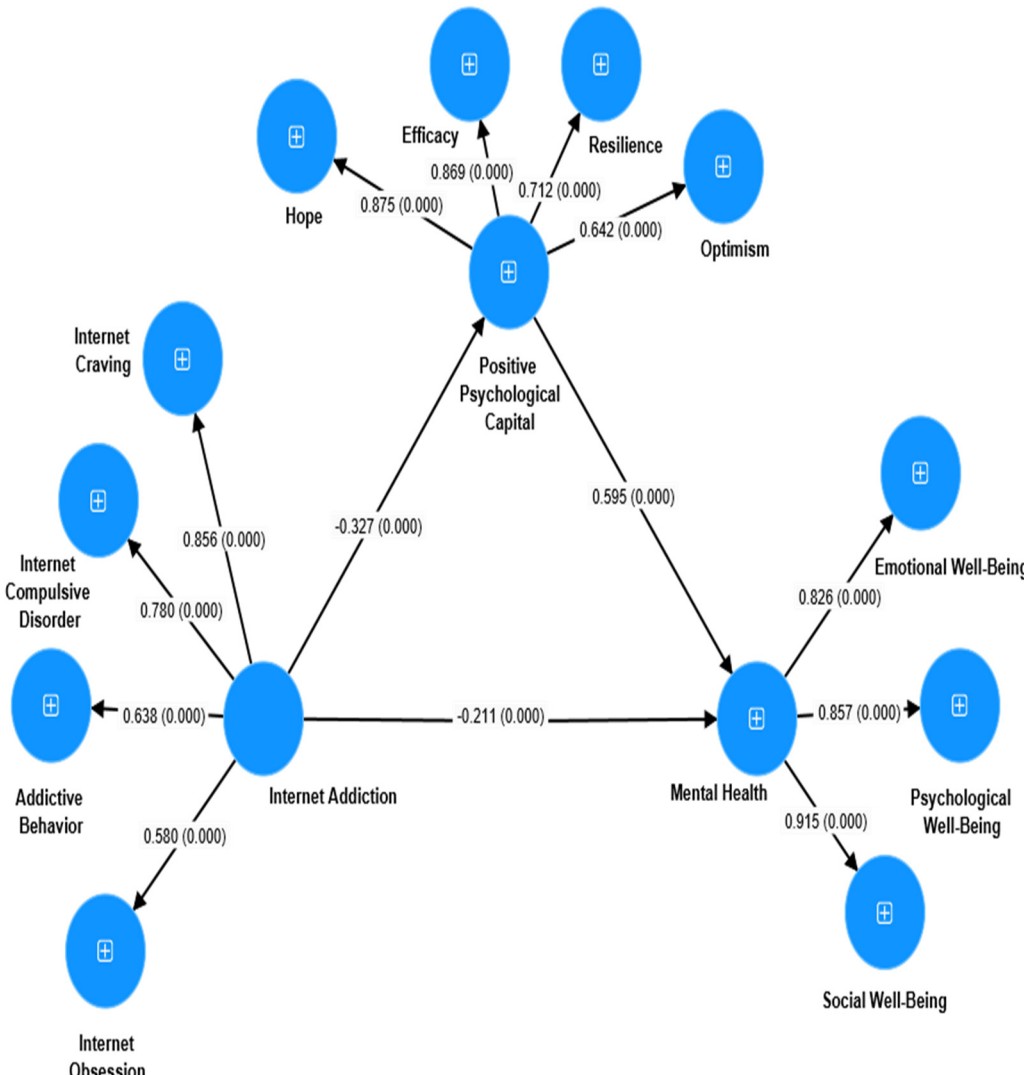

**Figure 5.** The Mediation Role of PsyCap between Internet addiction and mental health.

The results revealed that, in the tested model, internet addiction had a direct negative effect on PsyCap (β = −0.327, 95%CI [−0.414, −0.248], *p* = 0.001) and mental health (β = −0.211, 95%CI [−0.277, −0.140], *p* = 0.001). On the other hand, PsyCap demonstrated a significant and positive direct effect on mental health (β = 0.595, 95%CI [0.533, 0.658], *p* = 0.001). These findings supported Hypothesis 3.

Furthermore, in line with Hypothesis 4, this study discovered that PsyCap fully and significantly mediated the relationship between internet addiction and mental health (β = −0.195, 95%CI [−0.252, −0.146], *p* = 0.001).

## 4. Discussion

This study investigated two important issues: the psychometric properties of the three measures and the mediation analysis. The first hypothesis aimed at testing the psychometric properties of the three measures namely, the Psychological Capital Questionnaire (PCQ-24), the Internet Addiction Scale (IAS) and the Keyes' Mental Health Continuum-Short Form (MHC-SF) of the Amharic version in the Ethiopian context. Hence, the Ethiopian Amharic version of IAS, PCQ-24, and MHC-SF were checked for Cronbach alphas and composite reliability, convergent, discriminant and construct validity and proved that the three measures were reliable and valid. The AVE is greater than 0.05, and the AVE was less than MSV, squared correlation of the sub-constructs which we used with a sample of Ethiopian university students. In the second goal of this study as seen in Hypothesis

4 (see Figure 5) internet addiction (IA) predicts mental health (MH) as the dependent variable, with PsyCap serving as a mediating variable. This study also examined the relationship between IA and MH, with PsyCap as a mediating variable. The results showed a negative correlation between IA, PsyCap, and MH, indicating that higher levels of IA were associated with lower levels of PsyCap and MH, which supported the second hypothesis. However, PsyCap and MH were positively correlated, suggesting that higher levels of PsyCap were associated with a better MH. The negative prediction of internet addiction on both PsyCap and mental health highlights the detrimental effects of excessive internet use on the mental well-being of university students. Internet addiction is a growing concern in today's digitally connected world, with students being particularly vulnerable due to the extensive use of online platforms for study, communication, and entertainment.

These findings are consistent with the positive psychology theory of [31], which suggests that PsyCap and MH are linked. These findings are consistent with previous research [8,9], and support the third hypothesis that internet addiction negatively predicted mental health and a strong relationship was found between them. In addition, [8], found a relationship between social media overuse, psychological capital (i.e., positive psychological resources), and mental health outcomes and found that excessive social media use was associated with lower psychological capital and poorer mental health. In addition, a study found that higher levels of internet addiction resulted in lower psychological capital, poorer mental health [14], lower moral values and poor psychological well-being [6]. In addition, IA negatively influence psychological capital [13]. Studies by researchers [17,18], found that social networking site (SNS) addiction and higher levels of internet addiction were associated with higher levels of moral disengagement and poorer mental health. These findings have important implications for interventions and preventive measures aimed at addressing internet addiction and promoting mental health among university students. Efforts should focus on raising awareness about the potential risks of excessive internet use and providing strategies for healthy digital habits. Additionally, interventions should aim to enhance positive psychological resources through activities that foster self-efficacy, optimism, hope, and resilience. This finding aligns with previous research emphasizing the protective role of factors like hope, self-efficacy, optimism, and resilience in promoting mental well-being [8,9,17,26,55]. It suggests that interventions aimed at enhancing positive psychological resources can have a positive impact on mental health, particularly among university students who may face various stressors and challenges during their academic journey. University counselling services and educational programs can play a vital role in supporting students in developing coping skills, managing stress, and maintaining a healthy balance between online and offline activities.

The second main goal of this study was the mediating role of psychological capital in between internet addiction and mental health. There are several possible explanations for the observed mediation effect of positive psychological capital. First, excessive internet use can lead to diminished engagement in offline activities, such as social interactions, physical exercise, and face-to-face communication, which are crucial for developing and maintaining positive psychological resources. Second, internet addiction may contribute to feelings of isolation, loneliness, and decreased self-esteem, which can further deplete positive psychological capital and negatively impact mental health. Third, excessive internet use might disrupt sleep patterns, leading to fatigue and impaired cognitive functioning, which can directly influence positive psychological resources and mental health as well as indirectly affect the mental health of university students, support the fourth hypothesis.

This study suggests that positive psychological resources (hope, efficacy, resilience, and optimism) focusing on using positive psychology model, should lower internet addiction, increase healthy digital systems and improve the mental health of university students and their daily functioning. These positive psychological resources may lead to personal and organizational development and growth. This line of reasoning is supported by previous studies that have found PsyCap to be the best predictor of mental health and positive outcomes in students' lives, reducing stress, and fostering healthy digital

functioning [15,56,57]. Moreover, PsyCap has been identified as a preventive resource that can be used to improve mental health and lower internet addiction, and empirical evidence has shown a negative relationship between internet addiction and a positive relationship with mental health [19], which supports the hypotheses of this study.

It is important to note that this study focused specifically on university students, and further research is needed to generalize these findings to other populations. Additionally, this study relied on self-report measures, which may be subject to biases and limitations. Future research could employ longitudinal designs to examine the causal relationships between internet addiction, positive psychological capital, and mental health, as well as explore potential moderators and other mediating factors that may contribute to the observed associations. The findings underscore the importance of addressing internet addiction and promoting positive psychological capital as key factors in enhancing mental well-being and over all well-being [55–59]. These findings contribute to understanding the relationships between IA, PsyCap, and MH and highlight the importance of positive interventions for better mental health and healthy internet usage and positive PsyCap for promoting mental well-being among university students. By recognizing and addressing these issues, universities and relevant stakeholders can contribute to the overall holistic development and mental health of their students.

## 5. Conclusions

In conclusion, this study aimed to assess the psychometric suitability of three measures (IAS, PCQ-24, and MHC-SF) and investigate the mediating role of Psychological Capital (PsyCap) in the relationship between internet addiction (IA) and mental health (MH). The findings revealed several important insights. Firstly, the results demonstrated that internet addiction had a significant and negative direct effect on both PsyCap and MH. This indicates that higher levels of internet addiction among university students are associated with lower levels of positive psychological capital and poorer mental health outcomes.

Furthermore, this study revealed that PsyCap had a positive direct impact on MH. This suggests that positive psychological resources such as self-efficacy, optimism, hope, and resilience play a crucial role in promoting mental well-being among university students. Most importantly, this study found that PsyCap fully mediated the relationship between internet addiction and mental health. In other words, the detrimental effect of internet addiction on mental health is primarily driven by its impact on positive psychological resources. When university students have higher levels of internet addiction, it leads to lower positive psychological capital, which, in turn, contributes to poorer mental health outcomes. These findings underscore the significance of addressing internet addiction among university students. Not only does internet addiction directly affect mental health, but it also indirectly influences mental health through its impact on positive psychological capital. Therefore, interventions and preventive measures should focus on reducing internet addiction and promoting positive psychological resources to enhance mental well-being among university students.

In summary, this study highlights the importance of recognizing the relationship between internet addiction, positive psychological capital, and mental health in the context of university students. By addressing internet addiction and fostering positive psychological resources, we can improve the overall well-being of university students and promote their mental health.

**Author Contributions:** Conceptualization, G.T.Z., D.G.B., E.A, G.T. and T.S.; data collection, S.G., D.G.B. and T.S.; methodology, G.T.Z.; statistical analysis, G.T.Z.; material support, G.T, T.S., E.A. and S.G; writing—original draft preparation, G.T.Z. and D.G.B.; writing—review and editing, G.T.Z., G.T, E.A., D.G.B. and T.S. All authors have read and agreed to the published version of the manuscript.

**Funding:** This research received no external funding.

**Institutional Review Board Statement:** This article has been received ethical approval (Ref. No. 249-2023), January 21-2023 from the Institute of Teachers Education and Behavioural Sciences at Wollo University.

**Informed Consent Statement:** This study adhered to ethical standards outlined by the American Psychological Association (APA) and the National Association of Psychology regarding human subjects. Participants were fully informed about the study's purpose, procedures, and voluntary nature. They were assured of anonymity, and no identifying information or medical treatment was involved. Participants were aware of their right to withdraw from the study at any time. By initiating the survey, participants were considered to have read and accepted the informed consent.

**Data Availability Statement:** The corresponding authors hold the data sets generated and analyzed during this study and are willing to share them upon request.

**Conflicts of Interest:** The authors confirm that no conflicts of interest exist regarding this research, authorship, and publication of this article.

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
