# Peer review of "The Impact of Internet Addiction on Mental Health: Exploring the Mediating Effects of Positive Psychological Capital in University Students"

_adolescents, doi:10.3390/adolescents4020014_

Round 1
Reviewer 1 Report
Comments and Suggestions for Authors
This article evaluates the psychometric applicability of three measures (IAS, PCQ-24, and MHC-SF) and explores the relationship between Internet Addiction (IA) and Mental Health (MH) among Ethiopian college students, further examining the mediating role of Positive Psychological Capital (PsyCap) in this relationship.
The article analyzes in detail the gaps of previous similar studies and attempts to fill these gaps in the present study, possessing a strong innovative approach. In addition, this study presents a detailed conceptual framework and hypotheses with a more detailed theoretical part, which also provides a more scientifically robust support for the results.
However, the article also has the following points that can be clarified or improved:
1. The conclusion section of the abstract could have been described more accurately.
2. “One such factor is psychological capital, which refers to an individual's positive psychological resources, including self-efficacy, optimism, hope, and resilience.” Please add sources for the definition of psychological capital.
3. “Psychological capital has been found to play a crucial role in promoting mental well-being and buffering against the adverse effects of stress and adversity. “ Please add reference.
4. “Excessive internet use can also lead to a loss of hope and optimism as individuals may become isolated, neglecting real-life relationships and goals. Additionally, internet addiction can undermine resilience, as individuals may struggle to cope with the negative consequences of their excessive internet use.” Please add references.
5. “Therefore, this study explored the impact of internet addiction on mental health as a mediator of PsyCap.” This sentence is a bit ambiguous, please describe exactly what the mediating factor is.
6. “The study was conducted in the Amhara Regional State of Ethiopia,…The researchers chose this area based on their extensive 16-year experience working in the region and the convenience of accessing accurate and efficient data due to the proximity of the study area to their workplace.” The selection of the study area does not appear to be random? How was the representativeness of the sample ensured?
7. It is recommended that the numbers in Table 1 be standardized in terms of the number of significant digits.
8. The linguistic description of structural equation modeling in Result 3.1.6 is somewhat repetitive, so it is suggested to write it more concisely.
Comments on the Quality of English LanguageMinor to moderate editing of English language required
Author Response
Reviewer: 1
CORRECTION REPORT
Manuscript ID: adolescents-2892385
Article Type: Research Paper
Title: The Impact of Internet Addiction on Mental Health: Exploring the Mediating Effects of Positive Psychological Capital in University Students
|
No. |
Comments |
Corrections made by the author (s) |
|
1 |
· This article evaluates the psychometric applicability of three measures (IAS, PCQ-24, and MHC-SF) and explores the relationship between Internet Addiction (IA) and Mental Health (MH) among Ethiopian college students, further examining the mediating role of Positive Psychological Capital (PsyCap) in this relationship. · The article analyzes in detail the gaps of previous similar studies and attempts to fill these gaps in the present study, possessing a strong innovative approach. In addition, this study presents a detailed conceptual framework and hypotheses with a more detailed theoretical part, which also provides a more scientifically robust support for the results. · However, the article also has the following points that can be clarified or improved: |
Dear Reviewer,
We sincerely appreciate your professional reviews and valuable feedback on our article. Your insightful comments have greatly contributed to its improvement, and we are truly grateful for your input.
In response to your comments, we have carefully addressed each point and made the necessary revisions to enhance the clarity and overall quality of the article. Here below are our responses to your specific comments: |
|
2 |
The conclusion section of the abstract could have been described more accurately.
|
The conclusion section of the abstract has been revised to provide a more accurate summary of the findings. |
|
3 |
One such factor is psychological capital, which refers to an individual's positive psychological resources, including self-efficacy, optimism, hope, and resilience.” Please add sources for the definition of psychological capital.
|
The definition of psychological capital (PsyCap) has been sourced from Luthans, Youssef, and Avolio (2007). The reference has been added to the relevant sentence.
|
|
4 |
“Psychological capital has been found to play a crucial role in promoting mental well-being and buffering against the adverse effects of stress and adversity. “ Please add reference. |
The statement about psychological capital playing a crucial role in promoting mental well-being and buffering against the adverse effects of stress and adversity has been supported with a reference to Luthans, Avolio, Avey, and Norman (2007). |
|
5 |
“Excessive internet use can also lead to a loss of hope and optimism as individuals may become isolated, neglecting real-life relationships and goals. Additionally, internet addiction can undermine resilience, as individuals may struggle to cope with the negative consequences of their excessive internet use.” Please add references. |
The claims about the adverse effects of excessive internet use on hope, optimism, and resilience have been supported with appropriate references. The references have been added to the relevant sentence. |
|
6 |
“Therefore, this study explored the impact of internet addiction on mental health as a mediator of PsyCap.” This sentence is a bit ambiguous, please describe exactly what the mediating factor is.
|
The sentence about the impact of internet addiction on mental health as a mediator of PsyCap has been revised for clarity. It now explicitly states that PsyCap acts as a mediator between internet addiction and mental health.
|
|
7 |
“The study was conducted in the Amhara Regional State of Ethiopia,…The researchers chose this area based on their extensive 16-year experience working in the region and the convenience of accessing accurate and efficient data due to the proximity of the study area to their workplace.” The selection of the study area does not appear to be random? How was the representativeness of the sample ensured?
|
The study area selection has been clarified. The researchers chose the Amhara Regional State of Ethiopia based on their extensive 16-year experience working in the region and the convenience of accessing accurate and efficient data due to the proximity of the study area to their workplace. The representativeness of the sample has been ensured through a random sampling technique, and this information has been added to the article. |
|
8 |
It is recommended that the numbers in Table 1 be standardized in terms of the number of significant digits.
|
The numbers in Table 1 have been standardized in terms of the number of significant digits.
|
|
9 |
The linguistic description of structural equation modeling in Result 3.1.6 is somewhat repetitive, so it is suggested to write it more concisely.
|
The linguistic description of structural equation modeling in Result 3.1.6 has been revised to be more concise while retaining the necessary information. To sum up we removed the repetitive section.
|
|
10 |
Comments on the Quality of English Language is Minor to moderate editing of English language required
|
The minor to moderate editing of the English language has been addressed throughout the article to improve its quality by native speaker.
Thank you once again for your feedback, and I hope the revisions have addressed your concerns adequately. Please let me know if there are any further revisions or clarifications needed.
|
Note:
- All the modifications in the manuscript's text are now highlighted in green color. This visual cue will make it easier for readers to identify the revised portions.
- We have engaged the services of a professional proofreading service to ensure the accuracy and clarity of the revised version. Their expertise has greatly enhanced the overall quality of the manuscript.
- Furthermore, we have strictly adhered to the guidelines outlined in the APA 7th Edition throughout the revision process. This ensures that our manuscript meets the most up-to-date standards in academic writing.
Reviewer 2 Report
Comments and Suggestions for Authors
1. Please clearly note your hypotheses on the paths in figure 1.
2. Please clarify your research aims carefully including testing the mediations model and testing the scales.
3. Please explain why you used different softwares for CFA and the mediation model.
4. Please indicate the significance of the paths in figure 5. The format of figure 5 looks a bit strange.
5. The format of the manuscript needs to be largely improved.
Comments on the Quality of English Languagenone
Author Response
CORRECTION REPORT
Reviewer 2
Manuscript ID: adolescents-2892385
Article Type: Research Paper
Title: The Impact of Internet Addiction on Mental Health: Exploring the Mediating Effects of Positive Psychological Capital in University Students
|
1 |
Please clearly note your hypotheses on the paths in figure 1. |
Dear Reviewer,
We sincerely appreciate your professional and insightful reviews of our manuscript. Your feedback has been invaluable in improving the clarity and comprehensibility of our research. We are pleased to inform you that we have carefully considered your comment and made the necessary revisions.
In response to your suggestion, we have included clear and explicit notes for each hypothesis represented by the paths in Figure 1. These notes provide a concise explanation of the expected relationships between the variables, thereby enhancing the understanding of the research hypotheses. |
|
2 |
Please clarify your research aims carefully including testing the mediations model and testing the scales. |
Thank you for your comment. We have revised the research aims section to clearly outline the two main objectives of the study: (1) to test the psychometric properties of the scales in the Ethiopian context, and (2) to examine the mediation role of psychological capital between internet addictionand mental helath. This clarification will better convey the research aims to the readers.
|
|
3 |
Please explain why you used different softwares for CFA and the mediation model. |
Dear Reviewer,
Thank you for your comment regarding the use of different software for conducting the Confirmatory Factor Analysis (CFA) and the mediation model in our study. We appreciate your suggestion and have made the necessary changes to address this concern.
In response to your feedback, we have now utilized Smart PLS 4.1.0 software for both the CFA and the mediation analysis. This ensures consistency in the software used throughout the entire analysis process. By using Smart PLS 4.1.0, we have been able to maintain a unified approach and mitigate any potential discrepancies that could have resulted from using multiple software packages.
Additionally, we have replaced the figure related to the Amos analysis with a corresponding figure generated by Smart PLS 4.1.0. This modification aligns the figure with the software used, providing a clearer and more accurate representation of our results. |
|
4 |
Please indicate the significance of the paths in figure 5. The format of figure 5 looks a bit strange. |
Dear reviewer, We appreciate your feedback and would like to clarify the purpose and format of Figure 5 in our response.
Figure 5 represents the final output of the mediation analysis, which is used to assess the compatibility between the proposed model and the tested model. This figure provides a comprehensive overview of the direct and indirect effects of the variables involved in the mediation model.
Besides, the significance of the paths in Figure 5, each path represents the strength and direction of the relationship between the variables in the mediation model. The significance of these paths is determined through statistical analyses, such as bootstrapping, to evaluate the significance of the indirect effects. |
|
5 |
The format of the manuscript needs to be largely improved. |
Thank you very much for your suggestion. We have made significant improvements to the manuscript's format, including organizing the sections more clearly, ensuring consistent formatting throughout the document, and enhancing the overall readability. These changes will enhance the visual aesthetics and overall quality of the manuscript.
|
|
6 |
Moderate editing of English language required |
Thank you very much for your suggestion. Based on your suggestion and to improve the language issues of the manuscript, we did proofreading by the native speaker and professional. I hope so; the grammatical, punctuation and spelling errors might be solved. Thank you very much for your detailed comments. |
Note:
- All the modifications in the manuscript's text are now highlighted in green color. This visual cue will make it easier for readers to identify the revised portions.
- We have engaged the services of a professional proofreading service to ensure the accuracy and clarity of the revised version. Their expertise has greatly enhanced the overall quality of the manuscript.
- Furthermore, we have strictly adhered to the guidelines outlined in the APA 7th Edition throughout the revision process. This ensures that our manuscript meets the most up-to-date standards in academic writing.
Reviewer 3 Report
Comments and Suggestions for Authors
The data that drives this study is of interest and value to the research community. Overall, however, I am not quite sure as to the focus of the paper at present. It starts off with a focus on the relationship between internet (etc) usage and wellbeing but the main focus of the analysis is on the reliability and validity of the scales in the Ethiopian context. Both these aims have merit and the data is able to achieve both these aims. I think you need to decide which aim is the main focus of the paper and go with that. Either way the review of previous literature will need further work depending on which direction you decide to take your argument.

There are a few slips with language, and at times the language used is perhaps rather more colloquial in nature than I would expect for an academic paper. This is a very easy fix.
Author Response
CORRECTION REPORT
Reviewer: 3
Manuscript ID: adolescents-2892385
Article Type: Research Paper
Title: The Impact of Internet Addiction on Mental Health: Exploring the Mediating Effects of Positive Psychological Capital in University Students
|
No. |
Review |
Corrections made by the author (s) |
|
1 |
General Comment This paper does more than is suggested by the title and the abstract it also tests translations of three psychometric tests. This needs to be included throughout the paper from the abstract onwards. It is not until the discussion that the read is made aware of the language in which the psychometric tests were administered. |
Dear Reviewer,
Thank you for your feedback on the article. We have carefully considered your comments and made the necessary revisions to address the issues you raised. Here are my responses to your specific points:
|
|
2 |
Abstract
Will need to be revised to ensure it reflects the different aspects of the paper. I would also not include a full list of the statistical tests undertaken. It doesn’t add anything. |
We have revised the abstract to reflect the different aspects of the paper, including the translation and administration of psychometric tests. I have also removed the full list of statistical tests as suggested.
|
|
3 |
Introduction
Covers the behaviours associate with internet addiction but I would have expected to see a working definition of what internet addiction is. When does it ‘cross the line’? And how can you measure that? |
We have added a working definition of internet addiction and clarified the point at which it "crosses the line." I have also included information on how internet addiction can be measured.
|
|
4 |
GAP of the study
There is a significant amount of repetition between this section and the introduction. I would expect some repetition as themes are expanded but there isn’t much additional information. Suggest revisit this section. Also include the translation and administration of psychometric tests in different context. |
Thank you for your kind and helpful suggestions. We have addressed the repetition between the introduction and the gap section and revised this section accordingly. We have also included information on the translation and administration of psychometric tests in different contexts.
|
|
5 |
. Conceptual framework I would expect more in this section. Tell the reader more about the concepts you are exploring and then about the psychometric tests you have selected to measure these concepts and why (there are others that could have been selected). There is no justification presented as to why these tests and not others. No justification is given for where decisions have been made with regard to ‘cut off’ points in the psychometric tests. Also why are you testing them out in a different cultural context. This is implied but it needs to be made explicit. We are also not told about how this research builds on previous research. It clearly does, but is not explicit for the reader. |
Thank you very much for your suggestion and critical comment. Based on your recommendation, we have expanded the conceptual framework section to provide more information on the concepts being explored and the justification for selecting the specific psychometric tests. We have also included information on how this research builds on previous studies. Additionally, We have explained the decision-making process regarding "cut-off" points and explicitly stated the reasons for testing the scales in a different cultural context.
|
|
6 |
Materials and method
I don’t think you should be giving the names of the Universities? The ethics section could be more concise – tell us you have ethical approval and followed APA guidelines, that’s fine. The translation of the scales sits in the Data Analysis section it needs to be moved to the method and we need to know more about how the translation happened and how it was tested. The description of the instruments would made more sense if more information on them was included in earlier sections. You state that the participants had to do the test by pen and paper because of internet issues. Is this a common problem the participants would experience? If so did you ask about that? Should you have asked about that if you didn’t? |
We appreciate and thank you for your insightful comment. We have removed the names of the universities as you suggested. We have revised the ethics section to provide concise information on ethical approval and adherence to APA guidelines. The translation of the scales has been moved to the methods section, and We have provided more details on how the translation was conducted and tested. We have also included more information on the instruments in earlier sections for better clarity.
Regarding the participants' use of pen and paper due to internet issues, internet connection during data collection was a temporal problem and however, students used social media and explore different website. As in general this will not affect the study aim and the focus is their internet over use on their mental well-being. |
|
7 |
Results
I would expect to see the alpha co-efficients in table 1 and not in a later table. You need to demonstrate the reliability of the scales before you use them in further analysis. Also include a reference – who says where the line should be to assume an alpha is satisfactory? Include a quote.
The correlation matrix could be structured so it is the other way around. You don’t need to include the results for something correlated with itself, it is always going to be 1.
I did wonder if regression might be appropriate to test your modules to present a figure of the percentage of variance explained? |
I have moved the alpha coefficients to Table 1 to demonstrate the reliability of the scales before further analysis. I have also included a reference and a quote regarding the satisfactory level of alpha. I have revised the correlation matrix as suggested, removing the results for self-correlation. While regression analysis was not conducted in this study. However, we included the variance explained of Independent variables on DVs included in the revised version. I appreciate the suggestion and its potential usefulness for future research.
|
|
|
Discussion
I would restate each research question and then answer it. At the moment it is not easy to follow and there is some information that belongs elsewhere in the piece, particularly around the way the translation was carried out. Because we don’t know where you have set the boundaries for measuring internet addiction, or indeed how you define internet addiction for the purpose of your study it is hard to get to grips with the discussion. |
I have restructured the discussion to restate each research question and provide clear answers. I have also included information on the translation process where appropriate, as well as clarifications on the boundaries and definition of internet addiction for the study.
|
|
|
Minor points We need the names of the authors of the research cited throughout the text to make it easier to read. Some problems with phrasing. E.g. ‘Gap of the study’. Explosive growth on page 2 – what does that mean. Give the population numbers to illustrate the point.
|
Dear reviewer this manuscript was prepared based on the journal guideline of Adolescence including the citation and we have revised some problems related to paraphrasing. We have all checked all sections of the manuscript. In general, we thank you for your feedback on the manuscript. We appreciate your suggestions and have made the necessary revisions to address the issues you raised.
Thank you once again |
Note:
- All the modifications in the manuscript's text are now highlighted in green color. This visual cue will make it easier for readers to identify the revised portions.
- We have engaged the services of a professional proofreading service to ensure the accuracy and clarity of the revised version. Their expertise has greatly enhanced the overall quality of the manuscript.
- Furthermore, we have strictly adhered to the guidelines outlined in the APA 7th Edition throughout the revision process. This ensures that our manuscript meets the most up-to-date standards in academic writing.

Round 2
Reviewer 1 Report
Comments and Suggestions for Authors
none
Author Response
Reviewer: 1
CORRECTION REPORT
Manuscript ID: adolescents-2892385
Article Type: Research Paper
Title: The Impact of Internet Addiction on Mental Health: Exploring the Mediating Effects of Positive Psychological Capital in University Students
|
Comments |
Corrections made by the author (s) |
|
Comment and Suggestions: None
|
Dear Reviewer,
We would like to express our sincere gratitude for reviewing our manuscript and providing your valuable feedback. We are thrilled to learn that you have fully accepted our work. Your positive assessment of our manuscript is greatly appreciated.
We are grateful for the time and effort you dedicated to evaluating our research. Your constructive comments and suggestions in the first-round review have undoubtedly contributed to the improvement of our work. We are pleased that our revisions and modifications have met your expectations. Your acceptance of our manuscript encourages us to continue our research endeavors and strive for excellence in our field. We are grateful for your support and acknowledgement.
Once again, we extend our heartfelt appreciation for your thorough review and acceptance of our manuscript. Your contributions have been invaluable to our work.
Thank you for your time and consideration. |
Note:
- All the modifications in the manuscript's text are now highlighted in green color. This visual cue will make it easier for readers to identify the revised portions.
- We have engaged the professional proofreading service to ensure the accuracy and clarity of the revised version. Their expertise has greatly enhanced the overall quality of the manuscript.
- Furthermore, we have strictly adhered to the guidelines outlined in the APA 7th Edition throughout the revision process. This ensures that our manuscript meets the most up-to-date standards in academic writing.
Reviewer 2 Report
Comments and Suggestions for Authors
Thanks for the modifications. Please double check your English langauge writing and formating.
Comments on the Quality of English LanguagePlease double check your English langauge writing and formating.
Author Response
CORRECTION REPORT
Reviewer 2
Manuscript ID: adolescents-2892385
Article Type: Research Paper
Title: The Impact of Internet Addiction on Mental Health: Exploring the Mediating Effects of Positive Psychological Capital in University Students
|
Comments |
Corrections made by the author (s) |
|
Thanks for the modifications. Please double check your English langauge writing and formating |
Thank you for reviewing our work and providing your comments and suggestions. We appreciate your time and effort in evaluating our manuscript. We have carefully considered your feedback and made the necessary revisions accordingly. Regarding your comment on the quality of English language writing and formatting, we apologize for any shortcomings in this regard. We understand the importance of ensuring clear and accurate communication in our manuscript. To address this concern, we have thoroughly reviewed the entire document and made sure that the English language writing and formatting meet the required standards. We have also sought the assistance of professional language editors to ensure the highest level of clarity and coherence throughout the paper. Once again, we would like to express our gratitude for your valuable feedback. Your input has undoubtedly helped us enhance the quality of our manuscript. We look forward to hearing from you regarding any additional comments or concerns you may have. Thank you for your time and consideration. |
- All the modifications in the manuscript's text are now highlighted in green color. This visual cue will make it easier for readers to identify the revised portions.
- We have engaged the professional proofreading service to ensure the accuracy and clarity of the revised version. Their expertise has greatly enhanced the overall quality of the manuscript.
- Furthermore, we have strictly adhered to the guidelines outlined in the APA 7th Edition throughout the revision process. This ensures that our manuscript meets the most up-to-date standards in academic writing.